# Mechanical Properties of Bambusa Oldhamii and Yushania-Alpina Bamboo Fibres Reinforced Polypropylene Composites

**DOI:** 10.3390/polym14132733

**Published:** 2022-07-04

**Authors:** Yalew Dessalegn, Balkeshwar Singh, Aart W. van Vuure, Ali A. Rajhi, Alaauldeen A. Duhduh, Nazia Hossain, Gulam Mohammed Sayeed Ahmed

**Affiliations:** 1Program of Mechanical Design and Manufacturing Engineering, Department of Mechanical Engineering, Adama Science and Technology University, Adama 1888, Ethiopia; yalewdesu@yahoo.com (Y.D.); drgmsa786@gmail.com (G.M.S.A.); 2Department Materials Engineering, Campus Group T, Composite Materials Group, KU Leuven, Andreas Vesaliusstraat 13, 3000 Leuven, Belgium; aartwillem.vanuure@kuleuven.be; 3Department of Mechanical Engineering, College of Engineering, King Khalid University, Abha 61421, Saudi Arabia; arajhi@kku.edu.sa; 4Department of Mechanical Engineering Technology, CAIT, Jazan University, Prince Mohammed Street, P.O. Box 114, Jazan 45142, Saudi Arabia; adahduh@jazanu.edu.sa; 5School of Engineering, RMIT University, Melbourne, VIC 3001, Australia; bristy808.nh@gmail.com; 6Center of Excellence (COE) for Advanced Manufacturing Engineering, Adama Science and Technology University, ASTU, Adama 1888, Ethiopia

**Keywords:** age, bamboo species, harvesting seasons, impact strength, polypropylene composite, tensile strength

## Abstract

The current studies aim to measure the mechanical strength based on age, harvesting season and bamboo species in Ethiopia. The bamboo fibres are extracted using a roll milling machine, which was developed by the author. The age groups (1, 2 and 3 years), harvesting months (February and November), and bamboo species (Yushania alpina and Bambusa oldhamii) are the parameters of the current research studies. Prepregs and composites were produced from bamboo fibres and polypropylene. The mechanical properties of bamboo fibres and their composites in Ethiopia have not been investigated by researchers for the composite application so far. The tensile strength, Young’s modulus, and impact strength of injibara (Y. alpina) bamboo fibres reinforced PP composites from the ages of 1– 3 years old in November is 111 ± 9–125 ± 8 MPa, 15 ± 0.9–25 ± 0.72 GPa, and 47 ± 5 KJ/m^2^–57 ± 6 KJ/m^2^, whereas, in February, it is 86 ± 3.86–116 ± 10 MPa, 11 ± 0.71–23 ± 1.5 GPa, and 34 ± 4–52 ± 6 KJ/m^2^, respectively. Moreover, Kombolcha (B. oldhamii), bamboo fibres reinforced PP composites in November are 93 ± 7–111 ± 8 MPa, 7 ± 0.51–17 ± 2.56 GPa, and 39 ± 4–44 ± 5 KJ/m^2^, whereas, in February, it is 60 ± 5–104 ± 10 MPa, 12 ± 0.95–14 ± 0.92 GPa, and 26 ± 3 KJ/m^2^–38 ± 4 KJ/m^2^, respectively. Furthermore, Mekaneselam (Y. alpina) bamboo fibres reinforced PP composites in November are 99 ± 8–120 ± 11 MPa, 9 ± 0.82–16 ± 1.85 GPa, and 37 ± 4 KJ/m^2^–46 ± 5 KJ/m^2^, whereas, in February, it is 91 ± 8–110 ± 9 MPa, 8 ± 0.75–14 ± 1.86 GPa, and 34 ± 3 KJ/m^2^–40 ± 4 KJ/m^2^, respectively. At two years, November and Injibara bamboo have recorded the highest mechanical properties in the current research studies. Bamboo fiber strength in Ethiopia is comparable to the previous study of bamboo fibres and glass fibres used for composite materials in the automotive industry.

## 1. Introduction

Bamboo plants are a material used in many engineering applications and are grown in abundance in many tropical and subtropical regions of the world, especially in Asia. Moreover, it has the fastest growth rate of any plant and can provide excellent mechanical strength and rigidity because its cellulose fibers are aligned along its length [1,2]. With such benefits, bamboo fibers (BFs) appear to be an ideal potential replacement for glass fibers used in the production of fiber-reinforced composites [3].

Depending on the origin, fibers are categorized as either natural or synthetic. Nevertheless, each of these fibers has advantages and disadvantages. Although synthetic fiber-reinforced thermoplastic composites outperform natural fibers in comparison to mechanical strength, it is not eco-friendly [4,5,6]. The benefits of natural fibers instead of synthetic fibers in the production of fibres and their reinforced polymer composites have been driven by increased awareness of environmental protection and the need to use environmentally compatible materials. Advantages of natural fibers are the fact that they are cheap, light-weight, and biodegradable compared to synthetic fibers.

Hence, natural fibers utilized as reinforcing have become increasingly important research topics in thermoplastic-reinforced composites [7]. These composites have a wide variety of industrial uses in the fabrication of interior-automotive, furniture, and construction industries [8,9]. Fibers are used as the reinforcement and improve the strength of the composites, while polypropylene (PP) matrix maintains their desired location and orientation, as well as prevents them from environmental damage.

The preparation of PP composites is focused mainly on extrusion or hot pressing technology [10]. The main disadvantages of using plant fibers as reinforcements in a PP matrix are poor wettability and weak interfacial bonding because hydrophilic cellulose fibers are inherently incompatible and dispersible with hydrophobic PP. Tensile stresses within the fiber accumulate depending on the rank of fiber-matrix adhesion. Stress concentration reaches a certain high level beyond a certain level of the tensile strength (TS), which tends to result in deboning and fracture. PP is now a widely used polymer in many applications due to its numerous crucial and useful behavior like high heat distortion, high impact strength, transparency, dimensional stability, flame resistance, filling, reinforcing, and bending properties [11,12,13].

Impact tests are categorized into two types: low-velocity and high-velocity impact [14,15,16]. The first category includes the Charpy, Izod, and drop weight impact tests, while the second category includes the ballistics impact test. The Charpy and Izod impact test methods, also known as the pendulum method, are quick and simple to apply [17]. The function evaluates the quantities of energy absorbed by a notched or unnotched sample of a composite during fracture. The absorbed energy is a measure of the toughness of a composite and is used to study the ductile-brittle transition temperature [18].

The purpose of the research studies is to measure the strength of Ethiopian bamboo fibres and their reinforced PP composites based on the effect of age and harvesting season. The researched bamboo fibres were located in three regions of Ethiopia such as Injibara (Yushania Alpina), Kombolcha (Bambus oldhamii), and Mekaneselam (Yushania Alpina) bamboo. The properties of natural fibres are influenced by age, type of species, harvesting seasons, preparation of composites, and method of fibre extraction.

Currently, Ethiopian bamboo culms were used for the manufacturing of furniture, building houses, and fence. However, researchers have not investigated the properties of Ethiopian bamboo fibres and their reinforced PP composites for the composite application so far. The current research study performs the extraction of bamboo fibres, which was developed by the authors in the workshop.

A Prepreg was produced using a uniform distribution of fibres with polypropylene film using a hot press machine. Then, six layers of prepreg are used for bamboo fibre polypropylene composites production using a hot-pressing machine. The bamboo fiber was chosen as the reinforcement because it is a rich and diverse natural resource in Ethiopia and has overall mechanical strength comparable to glass fibers. Furthermore, the advantages of bamboo fibres are biodegradable, low processing energy, high specific strength, low cost, low densities, and renewable compared to glass fibres. Polypropylene was chosen as the matrix of the composite because it is a relatively cheap thermoplastic with reasonable mechanical properties.

## 2. Materials and Methods

### 2.1. Bamboo Plant Species

Bamboo plants were harvested in three regions of Amhara state, which are found in Ethiopia such as Injibara, Kombolcha, and Mekaneselam. The name of bamboo species based on the regions is Injibara (Y. alpina), Kombolcha (B. oldhamii), and Mekaneselam (Y. alpina). Figure 1 illustrates the geographical and climate condition of the research region for bamboo plants:

### 2.2. Sampling Technique

As indicated in Figure 2, the bamboo culm samples were harvested at 1–3 years old, in three regions, and in two seasons. Seasons in Ethiopia are categorized into 3 groups, such as rain, cold and dry seasons, based on climate conditions. The rain season (June–September) is an average annual temperature, humidity and annual rainfall are 19 °C, 68%, and 617 mm, respectively. The cold season (October–January) is an average annual temperature, humidity, and annual rainfall of 17 °C, 60%, and 149 mm, and the dry season (February–May), an average annual temperature, humidity, and annual rainfall of 24 °C, 55%, and 296 mm. The authors want to examine the harvesting season of bamboo species in the cold and dry seasons because of the low moisture content, and starch value of the bamboo culm exists in the cold and dry seasons. Moreover, three of each representative of bamboo culms are harvested at 1, 2 and 3 years old. Ages of the bamboo culm were known by experienced field personnel using the color of the culm, amount of the leaves and coverage of the sheath in the culm. The culm was later subdivided into three parts, such as the bottom, middle and top portions. The bamboo fiber was extracted at the middle position of the culm because the average diameter, thickness and larger internodal length are found at the middle positions.

### 2.3. Polypropylene (PP)

Polypropylene (PP) film with a density of 900 kg/m³ and 20 μm thickness was manufactured by Propex GmbH, Westfalen, Germany. The shaping, melting and coefficient of thermal expansion is 115.7 °C, 160.6 °C and (62.7–73.2) × 10^–6^/k, respectively. The Young’s modulus, strength, and strain to failure are 1.6–1.8 GPa, 55–65 MPa, and >300%, respectively.

### 2.4. Bamboo Fibres Extraction Using a Rolling Mill Machine

As shown in Figure 3a–c, Injibara, Kombolcha and M/selam are the classification of Bamboo culm cut from the nodes; then, the internodes were split along the longitudinal into strips of 30 mm. The strips were then rolled through the rolling mill’s two steel rods under low pressure and speed. One-third of the inner and outer parts of the culm are removed after enough rolling, then the middle section was combed using various sizes of comb for separate fibres. The obtained fibres were dried in the sun for 2 weeks, and their length ranged above 250 mm [19].

### 2.5. Tensile Stress

Tensile test samples were prepared based on the standard ASTM D3039 [20]. An Instron 4467 (Instron, Norwood MA, USA) machine with a load cell of 30 KN and a crosshead speed of 2 mm/min was applied. The distance between the two jaws was set at 150 mm and an extensometer with a 25 mm length of gauge was applied for measuring accurately the elongation of the composites. The samples were mechanically clamped using sandpaper in the grips to prevent slippage. The tensile test setup is shown in Figure 4. Before testing, all specimens were conditioned at room conditions (21 ± 2 °C and 50 ± 2% RH) for at least 24 h. The load and strain are registered during the complete test. At least five samples were tested for each bamboo species, age, as well as harvesting season. Sample dimensions of composites 250 × 10 × 2 mm were made with a target fibre volume fraction of 40%, which was calculated based on the weight and fibre density. Young’s modulus was calculated using the slope at the start of the stress-strain curve, which was between 0.1 and 0.3 percent of deformation [20].

### 2.6. Bamboo Fibre Prepreg Preparation

During bamboo fibre prepreg preparation, special care was taken to accurately align and evenly distribute the fibres in a unidirectional (UD) array. Plastic rubber is used for stabilizing the fibres uniformly, as seen in Figure 5a. Two layers of polypropylene film were attached to the fibres on both sides by using a hot press as seen in Figure 5b, c. The hot press adjusted a temperature of 170 °C, a compression pressure of 15 bar, and a waiting time of 10 s, enough time to pre-impregnate the bamboo technical fibres with the polymer Figure 5d. A Teflon^®^ sheet was applied in between the polymer and the hot plate during this operation to avoid the adhesion of the film to the plate. The average areal fibres density used in the prepregs was 250 ± 20 g/m^2^ [21,22,23,24].

### 2.7. Composite Production

The prepregs were dried in an oven at 65 °C for at least 72 h and subsequently placed in a desiccator to avoid moisture absorption, before composite production. Compression molding (Pinette hot press) was used to produce bamboo fiber reinforced polypropylene composites (BFRPPCs), as shown in Figure 6a–e. The prepregs are inserted in the mold, as shown in Figure 6a. The mold is covered with Teflon to prevent contamination with the hot press mold as shown in Figure 6b. The prepared mold inserted in the hot press machine to produce composite as shown in Figure 6c. The composites with the mold are removed from the hot press machine as shown in Figure 6d. The composite are prepared for further application as shown in Figure 6e. Unidirectional bamboo prepregs (6 layers) and thermoplastic films of PP were intercalated and placed into a cavity mold of 250 × 60 mm. The mold with the prepreg was inserted into a compression mold of the Pinette PEI, (Chalon sur Saone Cedex, France), hot press. The preheating and consolidation temperatures of the “Pinette” hot press machine were adjusted to170 °C, and 180 °C, respectively, with a waiting time of 5 min and a compression pressure of 20 bar. The fibre volume fraction (Vf) was targeted at 40% by weight measurements.

### 2.8. Sample Preparation for Testing

After the fabrication of the BFRPPCs (25 × 6 cm), samples with average dimensions of 250, 10 ± 0.2 and 1.85 ± 0.1 mm were cut using a low-speed banding saw machine (see Figure 7a,b. All specimens were placed at standard room temperatures (21 °C, 2 °C and 50% RH) for at least 24 h before testing.

### 2.9. Measurement of Impact Strength

Figure 8 shows the Izod impact tester. After breaking the specimen to a slightly lower height than a free swing, the pendulum continues to swing up. The specimen of the impact energy is determined by the energy lost by the pendulum.

## 3. Results and Discussion

### 3.1. Evaluation of Ultimate Tensile Strength

As shown in Figure 9, the back-calculated tensile strength of Injibara (Y. alpina), Kombolch (B. oldhamii), and Mekaneselam (Y. alpina) bamboo fibers is measured as a function of age and harvesting season. 

The highest and the lowest back-calculated tensile strength of Injibara (Y. alpina) bamboo fibres 386 ± 36 MPa and 326 ± 30 MPa were recorded in November, whereas, in February, 334 ± 23 MPa and 245 ± 30 MPa were recorded at the ages of 2 and 1 year old, respectively. The strength of 1 and 3-year-old Injibara (Y. alpina) bamboo fibres in November is 16% and 10% lower, whereas, in February, it is 27% and 13% lower than 2 years old, respectively. However, 3 years old in November is 6% higher, whereas, in February, it is 16% higher than 1 year old, respectively. Moreover, Injibara bamboo fibre in November was 13% higher compared to February. 

The highest and the lowest back-calculated tensile strength of Kombolcha (B. oldhamii) bamboo fibres 337 ± 32 MPa and 276 ± 25 MPa were recorded in November, whereas, in February, 328 ± 31 MPa and 270 ± 26 MPa were recorded at the ages of 2 and 1 year old, respectively. The strength of 1 and 3-year-old Kombolcha (B. oldhamii) bamboo fibres in November was 18% and 9% lower, whereas, in February, it was 18% and 1% lower than 2 years old, respectively. However, 3 years old in November is 10% higher, whereas, in February, it is 17% higher than 1 year old, respectively. Moreover, Kombolcha bamboo fibre in November was 3% higher compared to February.

The highest and the lowest back-calculated tensile strength of Mekaneselam (Y. alpina) bamboo fibres 379 ± 31 MPa and 293 ± 29 MPa were recorded in November, whereas, in February, 330 ± 32 MPa and 271 ± 25 MPa were recorded at the ages of 2 and 1 years old, respectively. The strength of 1 and 3-year-old Mekaneselam (Y. alpina) bamboo fibres in November was 23% and 19% lower, whereas, in February, it was 18% and 14% lower than 2 years old, respectively. However, 3 years old in November is 5% higher, whereas, in February, it is 6% higher than 1 year old, respectively. Moreover, Kombolcha bamboo fibre in November was 13% higher compared to February

Hence, the development of the cell wall of bamboo plants is great at a young age and stops after reaching maturity. The tensile strength of bamboo fibres increases from the age of 1 year to 2 years, then starts to decrease from the age of 2 years to 3 years old. The highest back-calculated tensile strength of bamboo fibers was recorded at the ages of 2 years and in November. From the highest to the lowest, back-calculated tensile strengths were recorded in Injibara (Y. alpina), Mekaneselam (Y. alpina) and Kombolch (B. oldhamii) bamboo fibers, respectively. The tensile strength of Mekaneselam and Kombolcha bamboo fibres in November was 2% and 13% lower, whereas, in February, it is 1% and 2% lower than Injibara bamboo fibres. However, Mekaneselam bamboo fibre in November is 11% higher, whereas, in February, it is 1% higher than Kombolcha bamboo fibres.

The current studies of back-calculated tensile strength of Injibara (Y. alpina), Kombolch (B. oldhamii), and Mekaneselam (Y. alpina) bamboo fibers are 51%, 52% and 57% lower than the previously reported by Wang et al.

### 3.2. Back-Calculated Young’s Modulus of Bamboo Fibres in Ethiopia

As shown in Figure 10, back-calculated Young’s moduli of Injibara (Y. alpina), Kombolch (B. oldhamii) and Mekaneselam (Y. alpina) bamboo fibers were measured with the influence of age and harvesting months. 

The highest and the lowest back-calculated Young’s moduli of Injibara (Y. alpina) bamboo fibres 43 ± 3 GPa and 28 ± 3 GPa were recorded in November, whereas, in February, 37 ± 4 GPa and 25 ± 3 GPa were recorded at the ages of 2 and 1 years old, respectively. The back-calculated Young’s moduli of 1 and 3-year-old Injibara (Y. alpina) bamboo fibres in November is 35% and 21% lower, whereas, in February, it is 32% and 5% lower than 2 years old, respectively. However, 3 years old in November is 18% higher, whereas, in February, it is 29% higher than 1 year old, respectively. Moreover, Injibara bamboo fibre in November was 14% higher compared to February.

The highest and the lowest back-calculated Young’s moduli of Kombolcha (B. oldhamii) bamboo fibres 35 ± 4 GPa and 29 ± 3 GPa were recorded at the age of 2 and 1 years old in November, whereas, in February, 25 ± 3 GPa and 18 ± 2 GPa were recorded at the ages of 2 and 1 years old, respectively. Kombolcha bamboo fibres in November were 29% higher in Young’s modulus compared to the value recorded in February. 

The back-calculated Young’s moduli of 1 and 3-year-old Kombolcha (B. oldhamii) bamboo fibres in November is 17% and 11% lower, whereas, in February, it is 28% and 20% lower than 2 years old, respectively. However, 3 years old in November is 10% higher, whereas, in February, it is 6% higher than 1 year old, respectively. Moreover, Kombolcha (B. oldhamii) bamboo fibre in November was 29% higher compared to February.

The highest and the lowest back-calculated Young’s moduli of Mekaneselam (Y. alpina) bamboo fibres 41 ± 5 GPa and 33 ± 3 GPa were recorded at the ages of 2 and 1 year old in November, whereas, in February, 32 ± 3 GPa and 25 ± 3 GPa were recorded at the ages of 2 and 1 year old, respectively. Mekaneselam bamboo fibres in November had a 22% higher Young’s modulus value compared to the value recorded in February. The calculated Young’s moduli of 1 and 3-year-old Mekaneselam (Y. alpina) bamboo fibre in November is 20% and 15% lower, whereas, in February, it is 22% and 6% lower than 2 years old, respectively. However, 3 years old in November is 6% higher, whereas, in February, it is 17% higher than 1 year old, respectively. Moreover, Mekaneselam (Y. alpina) bamboo fibre in November was 22% higher compared to February.

The Young’s modulus of Mekaneselam and Kombolcha bamboo fibre in November is 5% and 19% lower, whereas, in February, it is 14% and 32% lower than Injibara bamboo fibres. However, Mekaneselam bamboo fibre in November is 15% higher, whereas, in February, it is 20% higher than Kombolcha bamboo fibres.

The highest back-calculated Young’s modulus of bamboo fibers was recorded at the ages of 2 years old, and in November. From the highest to the lowest, back-calculated Young’s moduli was recorded in Injibara (Y. alpina), Mekaneselam (Y. alpina), and Kombolch (B. oldhamii) bamboo fibers, respectively. The current studies of Young’s modulus of Injibara, Kombolcha and Mekaneselam bamboo fibre polypropylene composites are 26%, 37% and 40% lower than the previously reported by Wang et al. [22].

### 3.3. Characterization of Strain to Failure in Ethiopia BFRPPCs

As shown in Figure 11, strain to failure of Injibara (Y. alpina), Kombolch (B. oldhamii), and Mekaneselam (Y. alpina) BFRPPCs was measured with the influence of age and harvesting months. 

The highest and the lowest strain of failure of Injibara (Y. alpina) BFRPPCs, 0.89 ± 0.06%, and 0.41 ± 0.04% were recorded in November, whereas, in February, 0.95 ± 0.06% and 0.79 ± 0.05% were recorded at the ages of 1 and 2 years old, respectively. However, in November, the strain to failure of 2 and 3-year-old Injibara (Y. alpina) BFRPPCs was 54% and 15% lower, respectively, than that of 1 year old. Three years old in November is 42% higher, whereas, in February, it is 29% higher than 2 years old, respectively. Moreover, Injibara (Y. alpina) BFRPPC in February was 6% higher compared to November.

The highest and the lowest strain of failure of Kombolcha (B. oldhamii) BFRPPCs 1.32 ± 0.10% and 0.63 ± 0.05% were recorded in November, whereas, in February, 1.21 ± 0.09% and 0.83 ± 0.06% were recorded at the ages of 1 and 2 years old, respectively. Kombolcha BFRPPCs in February had an 8% higher strain to failure compared to the value recorded in November. However, the strain to failure of 2 and 3-year-old Kombolcha (B. oldhamii) BFRPPCs in November is 52% and 37% lower, respectively, than that of 1 year old. Three years old in November is 24% higher, whereas, in February, it is 14% higher than 2 years old, respectively. Moreover, Kombolcha (B. oldhamii) BFRPPC in November was 8% higher compared to February.

The highest and lowest strain of failure of Mekaneselam (Y. alpina) BFRPPCs 0.87 ± 0.07% and 0.71 ± 0.06% were recorded at the age of 1 and 2 years old in November, whereas, in February, 0.98 ± 0.08%, and 0.76 ± 0.06% were recorded at the ages of 1 and 2 years old, respectively. Mekaneselam BFRPPCs in February had 11% higher compared to the value recorded in November. The strain to failure of 2 and 3-year-old Mekaneselam (Y. alpina) BFRPPCs in November is 18% and 7% lower, respectively, whereas, in February, it is 22% and 11% lower than a year old. Three years old in November, it is 12% higher, whereas, in February, it is 13% higher than 2 years old, respectively. Moreover, Mekaneselam (Y. alpina) BFRPPC in February was 11% higher compared to November.

The strain to failure of Injibara and Mekaneselam BFRPPCs in November was 33% and 34% lower, whereas, in February, they were 22% and 19% lower than Kombolcha BFRPPCs. However, Mekaneselam BFRPPC in November is 2% lower, whereas, in February, it is 3% higher than Injibara BFRPPCs. The highest strain to failure of BFRPPCs was recorded at the ages of 1 and in February as shown in Table 1. From the highest to the lowest, strains to failure were recorded in Kombolch (B. oldhamii), Mekaneselam (Y. alpina), and Injibara (Y. alpina) BFRPPCs, respectively.

### 3.4. Evaluation of Young’s Modulus Bamboo Fibre Polypropylene Composite

As indicated in Table 1, Young’s modulus of bamboo fibre polypropylene composite for Injibara (Y. alpina), Kombolch (B. oldhamii), and Mekaneselam (Y. alpina) BFRPPCs were measured using standard specimen preparation.

The highest Young’s moduli of Injibara bamboo fibre composite of 23 ± 1.5 GPa and 25 ± 2.21 GPa were recorded in the theoretical and experimental values of the harvesting month in February, whereas, in November, 25 ± 1.71 GPa and 26 ± 2.28 GPa are recorded, respectively. The efficiency factors of 92% and 96% are reached in February and November, respectively.

The Young’s moduli of 1 and 3-year-old Injibara (Y. alpina) bamboo fibre PP composite in November are 30% and 40% lower, whereas, in February, they are 52% and 48% lower than at 2 years old, respectively. However, 3 years old in November is a similar result recorded, whereas, in February, it is 8% higher than 1 year old, respectively. Moreover, Injibara (Y. alpina) bamboo fibre PP composite in November was 8% higher compared to February. Moreover, the highest theoretical and experimental values of Young’s modulus of Kombolcha bamboo fibres composite in February were 14 ± 0.92 GPa and 16 ± 0.94 GPa, whereas, in November, was 17 ± 2.56 GPa and 18 ± 1.12 GPa, respectively. Efficiency factors of 88% and 94% are recorded in February and November, respectively. 

The Young’s moduli of 1 and 3-year-old Kombolcha (B. oldhamii) bamboo fibre PP composites in November are 59% and 24% lower, whereas, in February, they were 14% and 7% lower than 2 years old, respectively. However, 3 years old in November is 46%, whereas, in February, it is 8% higher than 1 year old, respectively. Moreover, Kombolcha (B. oldhamii) bamboo fibre PP composites in November are 18% higher compared to February.

Furthermore, the highest theoretical and experimental values of Young’s modulus of Mekaneselam bamboo fibres composite were 17 ± 1.56 GPa and 18 ± 1.12 GPa in February, whereas, in November, were 16 ± 1.85 GPa and 19 ± 1.92 GPa, respectively. Efficiency factors of 94% and 84% were recorded in the harvesting months of February and November, respectively. The Young’s moduli of 1 and 3 years old of Mekaneselam (Y. alpina) bamboo fibres PP composites in November are 48% and 6% lower, whereas, in February, are 43% and 7% lower than 2 years old, respectively. However, 3 years old in November is 40%, whereas, in February, is 38% higher than 1 year old, respectively. Moreover, Mekaneselam (Y. alpina) bamboo fibre PP composites in November were 13% higher compared to February.

The Young’s moduli of Mekaneselam and Kombolcha bamboo fibres PP composites in November are 36% and 32% lower, whereas, in February, are 39% and 39% lower than Injibara bamboo fibre PP composites. However, Mekaneselam bamboo fibre PP composites are similar results to Kombolcha bamboo fibre PP composites.

In general, Young’s modulus of bamboo fibre polypropylene composite of the theoretical value was comparable to the experimental value. This indicates good fibre/matrix adhesion, possibly improved due to good fibre extraction and fiber special properties that promote mechanical interlocking.

The current studies of Young’s modulus of Injibara, Kombolcha, and Mekaneselam bamboo fibre polypropylene composites were higher than the previously reported by Chen et al. (3.4 GPa) [23], Cabrera et al. (1.3 GPa) [24], Shah et al. (3.75 GPa) [25], Ibrahim et al. (1.14 GPa) [26], Liu et al. (2.29 GPa) [27], Thwe et al. (2.15 GPa) [28], Samal et al. (1.25 GPa) [29], Chattopadhyay et al. (1.78 GPa) [30] and Keya et al. (4.96 GPa) [31].

### 3.5. Ultimate Tensile Strength of Bamboo Fibre Polypropylene Composites

As presented in Table 2, the highest and lowest tensile strength of Injibara bamboo fibre polypropylene composites 116 ± 10 MPa and 86 ± 4 MPa were recorded in February, whereas, in November, 125 ± 8 MPa and 111 ± 9 MPa were recorded, respectively. The strengths of 1 and 3 years old of Injibara (Y. alpina) bamboo fibre PP composites in November are 11% and 9% lower, whereas, in February, are 26% and 21% lower than 2 years old, respectively. However, 3 years old in November is 3% higher, whereas in February is 7% higher than 1 year old, respectively. Moreover, Injibara bamboo fibre PP composites in November were 7% higher compared to February.

Moreover, the highest and lowest tensile strength of Kombolcha bamboo fibre polypropylene composites 104 ± 10 MPa and 60 ± 5 MPa were recorded in February, whereas 111 ± 8 MPa and 93 ± 7 MPa were recorded in November, respectively. The strength of 1 and 3 years old of Kombolcha (B. oldhamii) bamboo fibres PP composites in November are 16% and 1% lower, whereas in February are 42% and 3% lower than 2 years old, respectively. However, 3 years old in November is 15% higher, whereas in February is 41% higher than 1 year old, respectively. Moreover, Kombolcha bamboo fibre PP composites in November were 6% higher compared to February.

Furthermore, the highest and lowest tensile strength of Mekaneselam bamboo fibre polypropylene composites 110 ± 9 MPa and 91 ± 8 MPa were recorded in February, whereas in November 120 ± 11 MPa and 99 ± 8 MPa were recorded, respectively. The strengths of 1 and 3 years old of Mekaneselam (Y. alpina) bamboo fibres PP composites in November are 18% and 16% lower, whereas in February are 17% and 15% lower than 2 years old, respectively. However, 3 years old in November is 2% higher, whereas in February is 3% higher than 1 year old, respectively. Moreover, Mekaneselam bamboo fibre PP composites in November were 8% higher compared to February.

The ultimate tensile strength of Mekaneselam and Kombolcha bamboo fibre PP composites in November are 4% and 11% lower, whereas in February are 5% and 10% lower than Injibara bamboo fibres. However, Mekaneselam bamboo fibre PP composites in November are 8% higher, whereas in February is 5% higher than Kombolcha bamboo fibres PP composites [32].

The current studies of tensile strength of Injibara (Y. alpina), Kombolcha (B. oldhamii), and Mekaneselam (Y. alpina) bamboo fibre polypropylene composites were higher than the previously reported by Chen et al. (36 MPa) [23], Cabrera et al. (5.4 MPa) [24], Shah et al. (16.25 MPa) [25], Ibrahim et al. (31.5 MPa) [26], Liu et al. (16.9 MPa) [27], Thwe et al. (35 MPa) [28], Samal et al. (40.5 MPa) [29], Chattopadhyay et al. (26.7 MPa) [33] and Keya et al. (62 MPa).

### 3.6. Measurement of Impact Strength of Bamboo Fibre Polypropylene Composite

As shown in Figure 12, the impact strength of Injibara, Kombolch and Mekaneselam bamboo fibre polypropylene composites was recorded with the influence of age and harvesting season. The highest and lowest values of impact strength of Injibara bamboo fibre polypropylene composites, 52 ± 6 KJ/m^2^ and 34 ± 4 KJ/m^2^, were measured at 2 and 1 years old in February, whereas in November 57 ± 6 KJ/m^2^ and 47 ± 5 KJ/m^2^ were recorded, respectively. The impact strengths of 1 and 3 years old of Injibara (Y. alpina) bamboo fibre PP composites in November are 18% and 12% lower, whereas in February are 35% and 25% lower than 2 years old, respectively. However, 3 years old in November is 6% higher, whereas in February is 13% higher than 1 year old, respectively. Moreover, Injibara bamboo fibre PP composites in November were 9% higher compared to February.

The highest and lowest values of impact strength of Kombolcha bamboo fibre polypropylene composites, 38 ± 4 KJ/m^2^ and 26 ± 3 KJ/m^2^, were measured at 2 and 1 years old in February, whereas in November 44 ± 5 KJ/m^2^ and 39 ± 3 KJ/m^2^ were recorded, respectively. Kombolcha bamboo fibre polypropylene composites in November had a 14% higher impact strength of bamboo fibre polypropylene composites value recorded compared to February. The impact strengths of 1 and 3 years old of Kombolcha (B. oldhamii) bamboo fibre PP composites in November are 11% and 7% lower, whereas in February are 32% and 26% lower than 2 years old, respectively. However, 3 years old in November is 5% higher, whereas in February is 8% higher than 1 year old, respectively. Moreover, Kombolcha bamboo fibre PP composites in November were 14% higher compared to February.

The highest and lowest values of impact strength of Mekaneselam bamboo fibre polypropylene composites, 40 ± 4 KJ/m^2^ and 34 ± 3 KJ/m^2^, were measured at 2 and 1 years old in February, whereas in November 46 ± 5 KJ/m^2^ and 37 ± 4 KJ/m^2^ were recorded, respectively. Mekaneselam bamboo fibre polypropylene composites in November had a 13% higher impact strength of bamboo fibre polypropylene composites value recorded compared to February. The impact strengths of 1 and 3 years old of Mekaneselam (Y. alpina) bamboo fibres PP composites in November are 20% and 15% lower, whereas in February are 15% and 10% lower than 2 years old, respectively. However, 3 years old in November is 5% higher, whereas in February is 6% higher than 1 year old, respectively. Moreover, Mekaneselam bamboo fibre PP composites in November were 13% higher compared to February.

The impact strengths of Mekaneselam and Kombolcha bamboo fibres in November are 19% and 23% lower, whereas in February are 23% and 27% lower than Injibara bamboo fibres. However, Mekaneselam bamboo fibres in November are 4% higher, whereas in February are 5% higher than Kombolcha bamboo fibres.

The current studies of the impact strength of Injibara, Kombolcha and Mekaneselam bamboo fibre polypropylene composites are higher than those previously reported by Cabrera et al. (10 KJ/m^2^) [24], Jain et al. (45 KJ/m^2^) [31], Liu et al. (3.7 KJ/m^2^) [27], Samal et al. (48 KJ/m^2^) [29], Chattopadhyay et al. (27 KJ/m^2^) [30] and Keya et al. (20 KJ/m^2^).

### 3.7. ANOVA Results of Tensile Test

As presented in Table 3, the ages of Injibara bamboo fibres are not statistically significant on the tensile strength, Young’s modulus, and strain to failure, whereas the harvesting season was not statistically significant on Young’s modulus, and strain to failure, but it is statistically significant on tensile strength difference between groups at *p* = 0.05 as determined by one-way ANOVA. 

The age of Kombolcha bamboo fibres is statistically significant on the tensile strength and Young’s modulus, but it did not strain to failure, whereas harvesting season was not statistically significant on the tensile strength, Young’s modulus, and strain to failure difference between groups at *p* = 0.05 as determined by one-way ANOVA. 

The age of Mekaneselam bamboo fibres is statistically significant, whereas the harvesting season was not statistically significant on the tensile strength, Young’s modulus and strain to failure difference between groups at *p* = 0.05 as determined by one-way ANOVA. 

### 3.8. ANOVA Results of Impact Test

As presented in Table 4, the age of Injibara, Kombolcha and Mekaneselam bamboo fibre polypropylene composite was statistically significant on the impact strength; moreover, the harvesting season of Injibara and Kombolcha bamboo fibre polypropylene composite was statistically significant, but Mekaneselam bamboo fibre polypropylene composite was not statistically significant on the impact strength difference between groups at *p* = 0.05 as determined by one-way ANOVA.

## 4. Conclusions

The tensile strength, Young’s modulus, strain to failure, and impact strength of bamboo fibres and their reinforced polypropylene composites in Ethiopia (Injibara, Kombolcha and Mekanselam) were measured based on the ages (1, 2 and 3 years) and harvesting months (February and November). The mechanical properties of Ethiopian bamboo fibres and their reinforced PP composites were found to increase from the ages of 1 year to 2 years, then decrease from the ages of 2 years to 3 years. The harvesting months of November have higher mechanical properties compared to February. The highest to the lowest mechanical properties of Ethiopian bamboo fibres and their composites are Injibara (Y. alpina), Mekaneselam (Y. alpina) and Kombolcha (B. oldhamii), respectively.

The back-calculated tensile strength and Young’s modulus ranging from 1 to 3 years old in February of Injibara (Y. alpina) bamboo fibres were recorded as 245 ± 30 MPa to 334 ± 23 MPa and 25 ± 3 GPa to 37 ± 4 GPa, Kombolcha (B. oldhamii) bamboo fibres were recorded as 270 ± 27 MPa to 328 ± 31 MPa and 18 ± 1 GPa to 25 ± 4 GPa, and Mekanselam (Y. alpina) bamboo fibres were recorded as 271 ± 26 MPa to 330 ± 32 MPa and 25 ± 2 GPa to 32 ± 4 GPa, whereas, in November, Injibara bamboo fibres were recorded as 326 ± 31 to 386 ± 36 MPa and 28 ± 3 to 43 ± 4 GPa, Kombolcha bamboo fibres were recorded as 276 ± 25 MPa to 337 ± 32 MPa and 29 ± 4 GPa to 35 ± 4 GPa, and Mekaneselam bamboo fibres were recorded 293 ± 28 MPa to 379 ± 30 MPa and 33 ± 2 GPa to 41 ± 5 GPa, respectively.

The ultimate tensile strength and Young’s modulus, with impact strength ranging from 1 to 3 years old in February of Injibara (Y. alpina) bamboo fibers reinforced polypropylene composites were recorded as 86 ± 3 MPa to 116 ± 10 MPa, 11 ± 0.71 GPa to 23 ± 1.5 GPa, and 34 ± 4 KJ/m^2^ to 52 ± 9 KJ/m^2^; Kombolcha bamboo fibres PP composites were recorded as 60 ± 5 MPa to 104 ± 10 MPa, 12 ± 0.95 GPa to 14 ± 0.92 GPa and 26 ± 4 KJ/m^2^ to 38 ± 4 KJ/m^2^, and Mekaneselam bamboo fibres PP composites were recorded as 91 ± 8 MPa to 110 ± 9 MPa, 8 ± 0.75 GPa to 14 ± 1.86 GPa and 34 ± 3 KJ/m^2^ to 40 ± 4 KJ/m^2^, whereas, in November, Injibara bamboo fibres PP composites were recorded as 111 ± 9 MPa to 125 ± 8 MPa, 15 ± 1.31 GPa to 25 ± 1.71 GPa and 47 ± 5 KJ/m^2^ to 57 ± 6 KJ/m^2^; Kombolcha bamboo fibres PP composites were recorded as 93 ± 7 MPa to 111 ± 7.75 MPa, 7 ± 0.51 GPa to 17 ± 2.56 GPa and 39 ± 5 KJ/m^2^ to 44 ± 5 KJ/m^2^, and Mekaneselam bamboo fibres PP composites were recorded as 99 ± 8 MPa to 120 ± 11 MPa, 9 ± 0.82 GPa to 16 ± 1.85 GPa and 37 ± 4 KJ/m^2^ to 46 ± 5 KJ/m^2^, respectively. Harvesting seasons influenced the properties of bamboo fibre reinforced PP composites. The Young’s moduli of Injibara (Y. alpina), kombolch (B. oldhamii) and mekaneselam (Y. alpina) bamboo fibre PP composites in November were 8%, 18% and 13% higher, whereas the tensile strengths were 7%, 6% and 8% higher, and impact strengths were 9%, 14% and 13% higher compared to February, respectively.

The current findings show that Ethiopian bamboo fibres and their composites are used for composite applications, which substitute the application area of glass fibres.

## Figures and Tables

**Figure 1 polymers-14-02733-f001:**
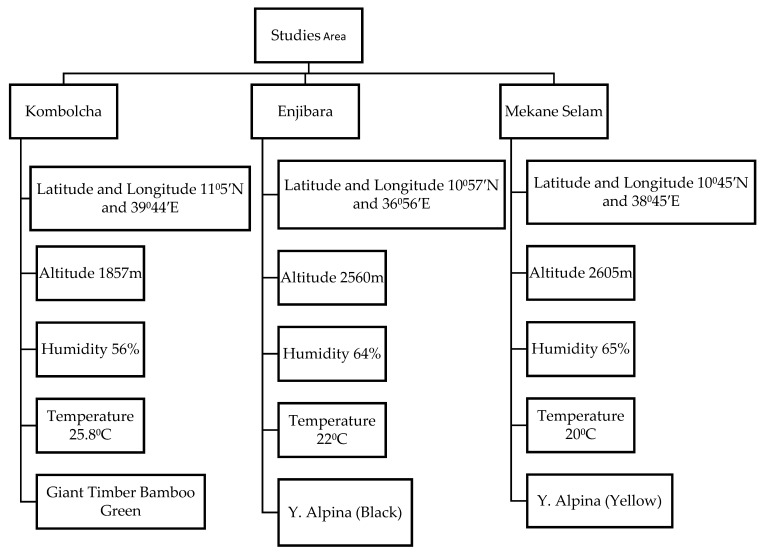
Description of the studied area.

**Figure 2 polymers-14-02733-f002:**
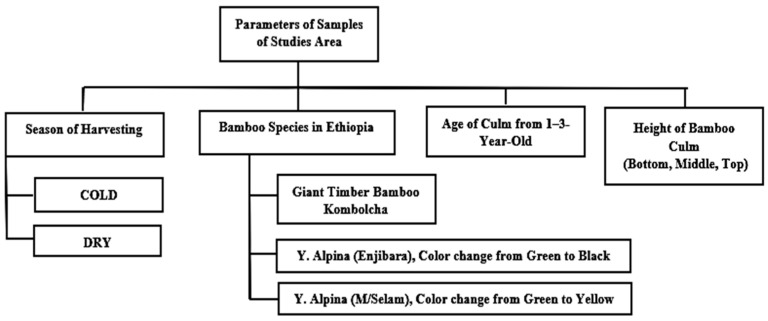
Sample taking techniques of the bamboo culm.

**Figure 3 polymers-14-02733-f003:**
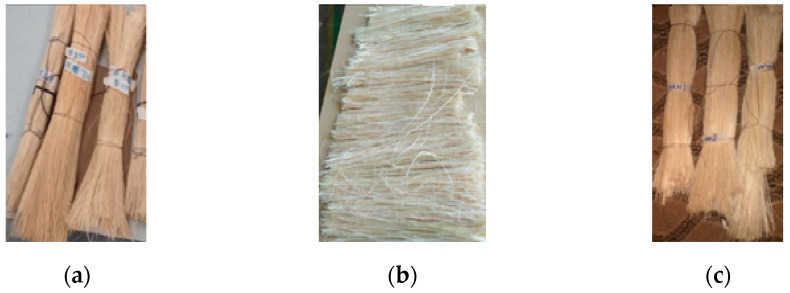
(**a**) Injibara, (**b**) Kombolcha, (**c**) M/selam.

**Figure 4 polymers-14-02733-f004:**
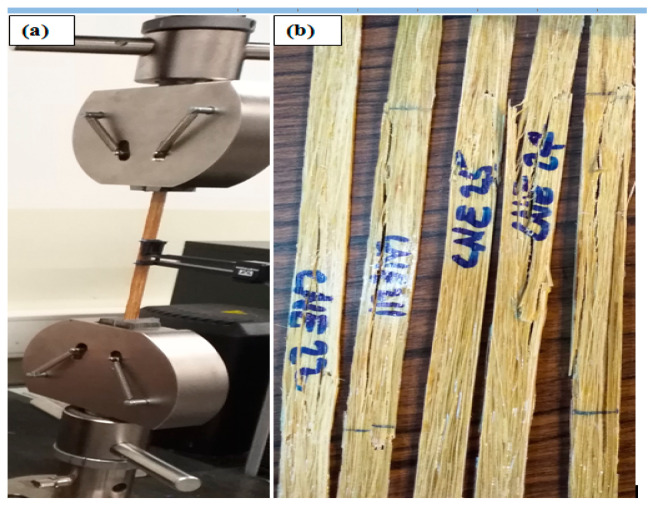
(**a**) Set-up of the tensile test machine; (**b**) mode of failure.

**Figure 5 polymers-14-02733-f005:**
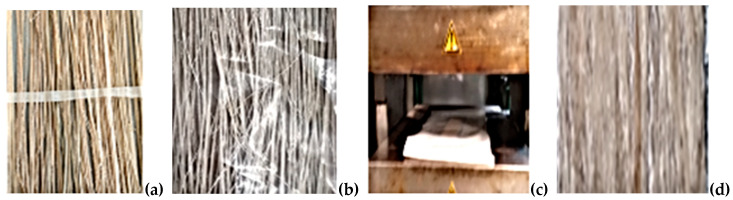
Preparation of the UD bamboo fibre prepregs with PP thermoplastic polymers. (**a**) Selection (**b**) Segregation (**c**) Drying (**d**) Soaking.

**Figure 6 polymers-14-02733-f006:**
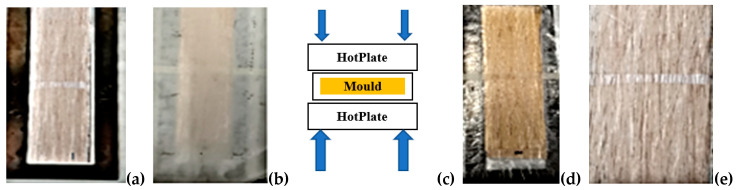
Preparation of bamboo fibre-PP composites: (**a**) bamboo fibre prepreg, (**b**) stacking sequence of prepregs, (**c**) compression molding, (**d**) composite in the mold, and (**e**) produced composite sample.

**Figure 7 polymers-14-02733-f007:**
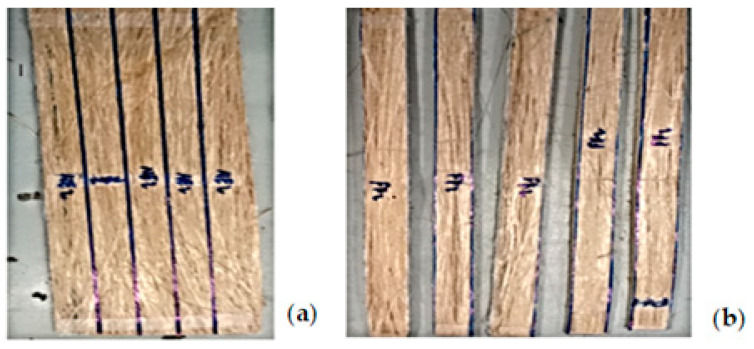
Samples view (**a**) samples layout for tensile testing, (**b**) after cutting using band saw.

**Figure 8 polymers-14-02733-f008:**
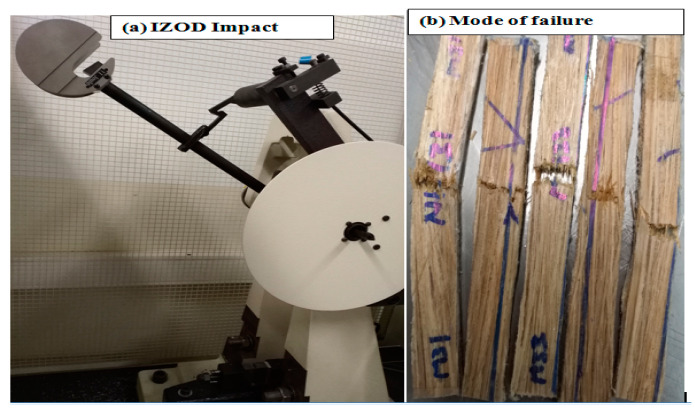
(**a**) Set up of Izod impact test, (**b**) mode of failure.

**Figure 9 polymers-14-02733-f009:**
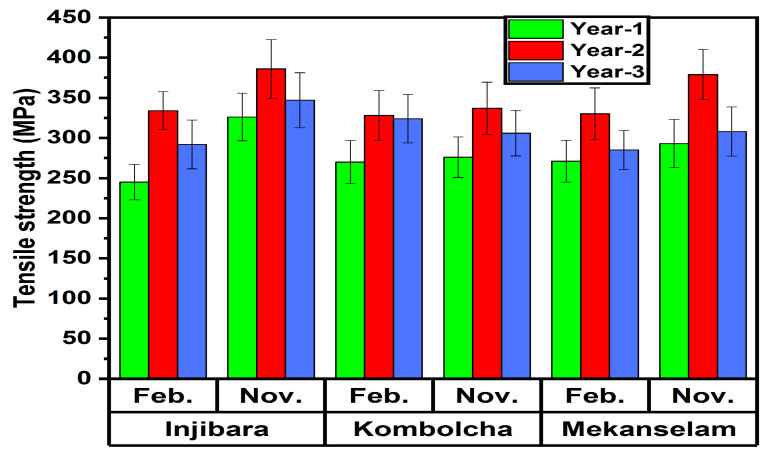
Back calculated tensile strength of the bamboo fibre in Ethiopia from BFRPPCs.

**Figure 10 polymers-14-02733-f010:**
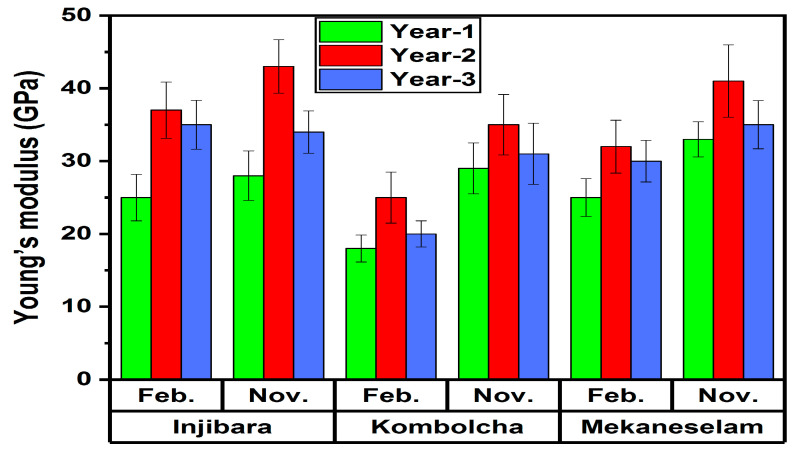
Back-calculated Young’s modulus of bamboo fibre in Ethiopia from BFRPPCs.

**Figure 11 polymers-14-02733-f011:**
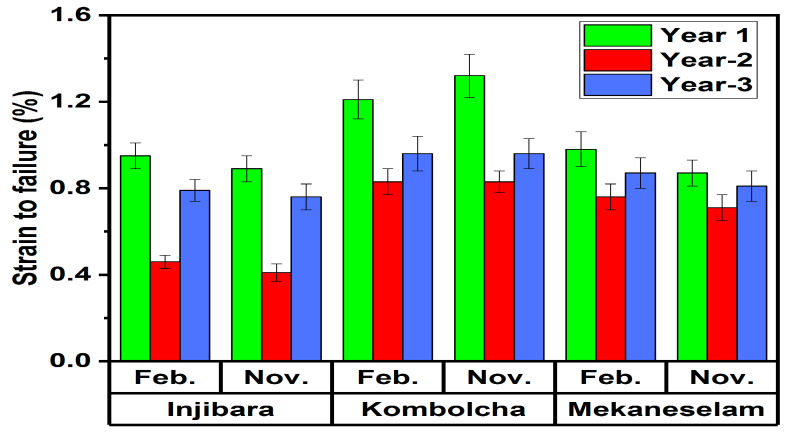
Strain to failure of bamboo composite in Ethiopia BFRPPCs.

**Figure 12 polymers-14-02733-f012:**
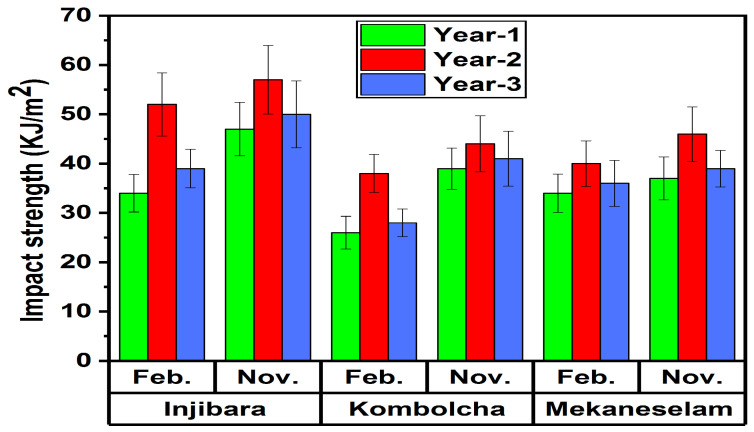
Impact strength of bamboo fibre polypropylene composite in Ethiopia.

**Table 1 polymers-14-02733-t001:** Standard methods of measuring unnotched impact strength.

Standard	DIN EN ISO 179-1: 2001–06 [21]
Specimen	The sample dimension of composite 80 × 10 × 4 mm was prepared with a target fibre volume fraction of 40%, which was calculated using the weight and fibre density.
Conditioning	The fibres were dried for at least 4 days at 60 °C before production and weighing. Pre-curing was done at 75 °C for 1 h, then post-curing for 1 h at 150 °C. The testing room shall be conditioned at 23 ± 2 °C and 50 ± 5% RH during testing.
Machine parameters	The Izod impact tester measures the toughness of a vertical cantilevered specimen hit by a swinging hammer. The Izod test exerts bending forces on the sample and measures the absorbed energy. Impact strength is expressed in [KJ/m^2^]. Impact testing was tested in the flatwise direction and striking velocity, fall angle and hammer weight were applied at 3.5 m/s, 124.34 degrees and 1.84 kg, respectively, used during the test.

**Table 2 polymers-14-02733-t002:** Experimental results of tensile strength of bamboo fibre polypropylene composites.

Species	Season	Ages	Exper. E Comp.	E Fibre	UTS Comp.	ε Comp.	UTS Fibre	Theor. E Comp.
		(Yr)	(GPa)	(Gpa)	(MPa)	(%)	(Mpa)	(Gpa)
INJ	February.	1	11 ± 0.71	25 ± 3	86 ± 3.86	0.95 ± 0.06	245 ± 30	13 ± 1.22
		2	23 ± 1.5	37 ± 4	116 ± 10	0.46 ± 0.03	334 ± 23	25 ± 2.21
		3	12 ± 2.9	35 ± 3	92 ± 7.11	0.79 ± 0.05	292 ± 30	20 ± 2.48
	November	1	15 ±1.18	28 ± 3	111 ± 9	0.89 ± 0.06	326 ± 31	13 ± 1.21
		2	25 ± 1.71	43 ± 4	125 ± 8	0.41 ± 0.04	386 ± 36	26 ± 2.28
		3	15 ± 1.31	34 ± 3	114 ± 6	0.76 ± 0.06	347 ± 34	14 ± 5.68
KOM	February.	1	12 ± 0.95	18 ± 1	60 ± 5	1.21 ± 0.09	270 ± 27	13 ± 1.51
		2	14 ± 0.92	25 ± 4	104 ± 10	0.83 ± 0.06	328 ± 31	16 ± 0.94
		3	13 ± 1.95	20 ± 2	101 ± 9	0.96 ± 0.0.08	324 ± 30	14 ± 1.54
	November	1	7 ± 0.51	29 ± 4	93 ± 7	1.32 ±0.1	276 ± 25	8 ± 0.62
		2	17 ± 2.56	35 ± 4	111 ± 7.75	0.63 ± 0.05	337 ± 32	18 ± 1.12
		3	13 ± 1.83	31 ± 4	110 ± 9	0.83 ± 0.07	306 ± 29	15 ± 1.25
MES	February.	1	8 ± 0.75	25 ± 2	91 ± 8	0.98 ± 0.08	271 ± 26	9 ± 0.85
		2	14 ± 1.86	32 ± 4	110 ± 9	0.76 ± 0.06	330 ± 32	16 ± 1.92
		3	13 ± 1.65	30 ± 2	94 ± 6	0.87 ± 0.07	285 ± 24	14 ± 1.25
	November	1	9 ± 0.82	33 ± 2	99 ± 8	0.87 ± 0.06	293 ± 28	10 ± 0.82
		2	16 ± 1.85	41 ± 5	120 ± 11	0.71 ± 0.06	379 ± 31	19 ± 1.92
		3	15 ± 1.72	35 ± 3	101 ± 8	0.81 ± 0.07	308 ± 30	17 ± 1.55

Note: INJ—Injibara, KOM—Kombolcha, MES—Mekaneselam, Exper.—Experimental, Theor. Theoretical.

**Table 3 polymers-14-02733-t003:** ANOVA test results of tensile test on the effect of ages and harvesting seasons.

	Injibara			Kombolcha			Mekaneselam		
Effect of Ages	TS	E	ε	TS	E	ε	TS	E	ε
*p*-value	0.7858	0.2961	0.2653	0.0372	0.0264	0.2237	0.0109	0.0000	0.0000
f-ratio	0.24	1.27	1.39	3.73	4.17	1.58	5.36	27.2	17.22
Signif.(*p* < 0.05)	No	No	No	Yes	Yes	No	Yes	Yes	Yes
Effect of season	TS	E	ε	TS	E	ε	TS	E	ε
*p*-value	0.0303	0.8136	0.1467	0.0632	0.8509	0.1309	0.1734	0.2135	0.1914
f-ratio	5.21	0.06	2.23	3.74	0.04	2.42	1.59	1.62	1.79
Signif. (*p* < 0.05)	Yes	No	No	No	No	No	No	No	No

Note: TS—Tensile strength, E—Young’s modulus, and ε-strain to failure.

**Table 4 polymers-14-02733-t004:** ANOVA test results of impact strength on the effect of ages and harvesting seasons.

	Injibara			Kombolcha			Mekaneselam		
Impact Strength	*p*-Value	f-Ratio	Sign.	*p*-Value	f-Ratio	Sign.	*p*-Value	f-Ratio	Sign.
Effect of Age	0.0407	5.71	Yes	0.0248	8.91	Yes	0. 0345	5.25	Yes
Effect of season	0.0159	6.45	Yes	0.0004	15.71	Yes	0.3887	0.76	No

## Data Availability

Not applicable.

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
