# Peer review of "Mechanical Properties of Bambusa Oldhamii and Yushania-Alpina Bamboo Fibres Reinforced Polypropylene Composites"

_polymers, 2022, doi:10.3390/polym14132733_

Round 1

Reviewer 1 Report

Dear,

The authors explored the development of composites using bamboo fiber. Discussions should be improved, as well as correlating properties with fiber type. Some suggestions are reported below:

1°) Please correct the term "Poly-propylene" in the manuscript. Correct to polypropylene.

2°) Abstract. Add quantitative data from experimental results.

3°) Please arrange the introduction. I suggest organizing in paragraphs.

4°) Page 2. Line 55-57. The paragraph connection makes no sense. The authors talk about polypropylene without previously approaching the most used thermoplastic matrices. Please clarify the novelty of the manuscript.

5°) Why didn't the authors add detailed characterizations of the fibers? Did you do thermogravimetry? Scanning electron microscopy (SEM)?

6°) Moisture absorption testing is very important to be investigated in composites. Did the authors not perform the analysis?

7°) Page 7. Line 194-216. Authors should further discuss the observed differences in tensile strength. How varying harvest time affects fiber structure. In view of this, how was the influence on the property.

8°) Page 8. Line 246-264. Again, the authors must correlate the variation in fiber harvest time with the results.

9°) Discussions should be improved, the authors only report the experimental data. Authors should try to explain the influence of harvest time and fiber type on properties.

10°) Authors should add scanning electron microscopy of composites. Then, correlate the properties obtained with the morphology.

Author Response

Title: Mechanical Properties of Bambusa Oldhamii and Yushania-Alpina Bamboo Fibres Reinforced Polypropylene Composites

MATERIALS-ID- 1775340

The authors are highly grateful to the reviewers for their constructive comments, which have helped to enhance the quality of the manuscript. Author's sincere effort has been put to revise the manuscript according to the reviewer comments. The details of revisions made in response to the comments are summarized in the following Table. The revisions in the manuscript are shown in YELLOW colored texts.

REVIEWER-COMMENT-1

AUTHOR RESPONSE

1# Please correct the term "Poly-propylene" in the manuscript. Correct to polypropylene.

Reply: Thank you for your valuable comments. As per your comments, the authors corrected the term of polypropylene

2# Abstract. Add quantitative data from experimental results.

Reply: Thank you, we agree with the valuable comment by the reviewer to improve the quality of the manuscript:

We add quantitative data from the experimental results.

3# Please arrange the introduction. I suggest organizing in paragraphs

Reply: The authors appreciate the reviewer for your valuable comments on the introduction parts.

We arranged and organized the introduction parts in to paragraph.

4# Page 2. Line 55-57. The paragraph connection makes no sense. The authors talk about polypropylene without previously approaching the most used thermoplastic matrices. Please clarify the novelty of the manuscript.

Reply: Thank you for your requisition of clarity on the novelty of the manuscript.

As we know, natural fibres did not have manufacturing data like synthetic fibres due to they are influenced by many parameters like age, harvesting seasons, climate condition and altitude of the regions,  extraction methods, type of species, manufacturing and preparation methods of composite, etc..

The properties of Ethiopian bamboo fibres and their composites have not been investigated by researchers so far. The aim of the current research studies is to measure the strength of Ethiopian bamboo fibres and their polypropylene composites based on age, harvesting season, and region using authors developed extraction machine. 

5# Why didn't the authors add detailed characterizations of the fibers? Did you do thermogravimetry? Scanning electron microscopy (SEM)?

Reply: Thank you for your questions,

I have been working my PhD dissertation on the full characterization of Ethiopian bamboo species like morphology, physical, chemical, and mechanical properties based on age and harvesting seasons. When I incorporate all in the manuscript, the manuscript may be difficult to manage in size.

The aim of the current research manuscript are focused on the mechanical properties of bamboo species based on ages (1,2, and 3 years) and harvesting months (February and November) to reduced the bulk of the manuscript.

Authors did not need to incorporate the topics of TG and SEM on the current research manuscript due to the scope of the manuscript

6# Moisture absorption testing is very important to be investigated in composites. Did the authors not perform the analysis?

Reply: Thank you for your questions,

The application area of the composites are limited by the properties of composites. Moisture absorption is one of the criteria that affect the applicability area of the composites. we did not need to study  the behavior moisture absorption when composites are used for out of contamination of moisture. Authors have investigated the hydrothermal properties of composites, but we did not incorporated for the current research topic due to scope of the manuscript.

7# Page 7. Line 194-216. Authors should further discuss the observed differences in tensile strength. How varying harvest time affects fiber structure. In view of this, how was the influence on the property.

Reply: Authors sincerely appreciate the reviewer for the valuable comment.

As per your comments, we discussed the observed differences in tensile strength.

One of the properties of natural fibres are influenced by harvesting seasons. The seasons in Ethiopia are categorized into cold, dry and rainy. Cold and dry  seasons are used for  the current research work. The moisture content of bamboo plants are higher when harvested in cold seasons. The extraction process are influenced by the amount of moisture content in the bamboo plant due to easily crushed and removed the long fibres without damage. If the moisture content is low, the fibres is  damaged during extraction. Hence the properties of bamboo fibres are affected directly by the extraction process and indirectly by the moisture content.  

8# Page 8. Line 246-264. Again, the authors must correlate the variation in fiber harvest time with the results.

Reply: we agree with the valuable comment by the reviewer as per the comments, the authors correlated harvest time with the results

9# Discussions should be improved, the authors only report the experimental data. Authors should try to explain the influence of harvest time and fiber type on properties.

Reply: We sincerely appreciate the reviewer for the valuable comment, which have improved the quality of our article.

As per your comments, the authors discussed more on the influence of harvest time and fibres properties.

10# Authors should add scanning electron microscopy of composites. Then, correlate the properties obtained with the morphology.

Reply: Thank for your suggestion.

Scanning electron microscopy (SEM ) used to study the morphology of fibre to matrix composites before and after mechanical test that characterized the behavior of failure of the fibres and the matrix.

SEM is very important to study the type of failure behavior of the composites.

The type of failure behavior is out of the scope in the current research work.

Reviewer 2 Report

In this study, authors prepared bamboo reinforced polypropylene composites and investigated their mechanical properties. I urge the authors to address the given comments.

1. There are many grammatical mistakes in manuscript. It should be thoroughly revised.

 2. I's need to check the thermal stability of prepared composites.

3. Hardness of prepared composites should also be evaluated.

4. Cite some latest papers of the respective journal.

Author Response

Title: Mechanical Properties of Bambusa Oldhamii and Yushania-Alpina Bamboo Fibres Reinforced Polypropylene Composites

MATERIALS-ID- 1775340

The authors are highly grateful to the reviewers for their constructive comments, which have helped to enhance the quality of the manuscript. Author's sincere effort has been put to revise the manuscript according to the reviewer comments. The details of revisions made in response to the comments are summarized in the following Table. The revisions in the manuscript are shown in YELLOW colored texts.

REVIEWER-COMMENT-1

AUTHOR RESPONSE

1# There are many grammatical mistakes in manuscript. It should be thoroughly revised.

Reply: The authors appreciate the reviewer for your valuable comments on many grammatical errors. The authors have edited and revised the manuscript to improve the quality.

2# I's need to check the thermal stability of prepared composites.

Reply: Thank you for your suggestion.

The application area of the composites are limited by the properties of composites. Thermal property is one of the criteria that affect the applicability area of the composites. We did not need to study  the behavior thermal properties when composites did not subject to contamination of thermal. Authors have investigated the hydrothermal properties of composites, but we did not incorporated for the current research topic due to scope and aim of the manuscript.

3# Hardness of prepared composites should also be evaluated.

Reply: Thank you for your suggestion. The response of this comments are the same as authors respond in number 2#. The application area of the composites are determined by the properties of the composites. Hardness of the material are useful in the application area of blade, cutter and movable parts on the guide which have higher rubbing and surface contact when move one over the other.

The aim of the current research work  measured the mechanical properties ( ultimate tensile strength and impact strength) of Ethiopian bamboo fibres and their composites based on harvesting age and seasons. Hardness of composite materials is out of scope in the current research work.

4# Cite some latest papers of the respective journal.

Reply: We agree with the valuable comment by reviewer to improve the quality of the manuscript.

As per your comments, Authors added some latest papers from the respective journal.

Round 2

Reviewer 1 Report

The authors answered the questions satisfactorily. In view of this, the manuscript has been improved in quality.

Reviewer 2 Report

Authors have addressed the comments accordingly.